# Metabolome Profiling and Predictive Modeling of Dark Green Leaf Trait in Bunching Onion Varieties

**DOI:** 10.3390/metabo15040226

**Published:** 2025-03-26

**Authors:** Tetsuya Nakajima, Mari Kobayashi, Masato Fuji, Kouei Fujii, Mostafa Abdelrahman, Yasumasa Matsuoka, Jun’ichi Mano, Muneo Sato, Masami Yokota Hirai, Naoki Yamauchi, Masayoshi Shigyo

**Affiliations:** 1Laboratory of Vegetable Crop Science, Division of Life Science, Graduate School of Sciences and Technology for Innovation, Yamaguchi University, Yamaguchi 753-8515, Japan; 2Laboratory of Vegetable Crop Science, Faculty of Agriculture, Yamaguchi University, Yamaguchi 753-8515, Japan; 3Yamaguchi Prefectural Agriculture and Forestry General Technology Center, Mure 10318, Hofu 747-0004, Japan; 4Center of Biotechnology and Genomics, Texas Tech University, Lubbock, TX 49409, USA; 5Advanced Technology Institute, Yamaguchi University, Ube 755-8611, Japan; 6Science Research Center, Organization of Research Initiatives, Yamaguchi University, Yoshida 1677-1, Yamaguchi 753-8515, Japan; 7RIKEN Center for Sustainable Resource Science, 1-7-22 Suehiro-cho, Tsurumi-ku, Yokohama 230-0045, Japan; 8Department of Applied Biosciences, Graduate School of Bioagricultural Sciences, Nagoya University, Nagoya 464-8601, Japan

**Keywords:** *Allium fistulosum*, dark green color, metabolite profiling, machine learning, pigment compounds, pheophytin *a*

## Abstract

**Background:** The dark green coloration of bunching onion leaf blades is a key determinant of market value, nutritional quality, and visual appeal. This trait is regulated by a complex network of pigment interactions, which not only determine coloration but also serve as critical indicators of plant growth dynamics and stress responses. This study aimed to elucidate the mechanisms regulating the dark green trait and develop a predictive model for accurately assessing pigment composition. These advancements enable the efficient selection of dark green varieties and facilitate the establishment of optimal growth environments through plant growth monitoring. **Methods:** Seven varieties and lines of heat-tolerant bunching onions were analyzed, including two commercial $F_{1}$ cultivars, along with two purebred varieties and three $F_{1}$ hybrid lines bred in Yamaguchi Prefecture. The analysis was conducted on visible spectral reflectance data (400–700 nm at 20 nm intervals) and pigment compounds (chlorophyll *a*, chlorophyll *b* and pheophytin *a*, lutein, and β-carotene), whereas primary and secondary metabolites were assessed by using widely targeted metabolomics. In addition, a random forest regression model was constructed by using spectral reflectance data and pigment compound contents. **Results:** Principal component analysis based on spectral reflectance data and the comparative profiling of 186 metabolites revealed characteristic metabolite accumulation associated with each green color pattern. The “green” group showed greater accumulation of sugars, the “gray green” group was characterized by the accumulation of phenolic compounds, and the “dark green” group exhibited accumulation of cyanidins. These metabolites are suggested to accumulate in response to environmental stress, and these differences are likely to influence green coloration traits. Furthermore, among the regression models for estimating pigment compound contents, the one for chlorophyll *a* content achieved high accuracy, with an $R^{2}$ value of 0.88 in the test dataset and 0.78 in Leave-One-Out Cross-Validation, demonstrating its potential for practical application in trait evaluation. However, since the regression model developed in this study is based on data obtained from greenhouse conditions, it is necessary to incorporate field trial results and reconstruct the model to enhance its adaptability. **Conclusions:** This study revealed that cyanidin is involved in the characteristics of dark green varieties. Additionally, it was demonstrated that chlorophyll *a* can be predicted using visible spectral reflectance. These findings suggest the potential for developing markers for the dark green trait, selecting high-pigment-accumulating varieties, and facilitating the simple real-time diagnosis of plant growth conditions and stress status, thereby enabling the establishment of optimal environmental conditions. Future studies will aim to elucidate the genetic factors regulating pigment accumulation, facilitating the breeding of dark green varieties with enhanced coloration traits for summer cultivation.

## 1. Introduction

The bunching onion (*Allium fistulosum* L.), also known as Welsh onion, green onion, spring onion, or scallion, is widely distributed from Siberia to tropical Asia, particularly in East Asia, where numerous varieties have adapted to diverse local environmental conditions [1,2]. It is consumed year round, with its green color and distinctive flavor being essential components of various dishes [3,4,5]. The intensity of leaf color is a crucial trait directly linked to market value, nutritional content, and visual appeal. Furthermore, due to the impact of recent climate changes on the growth of bunching onions during summer, seed companies are developing varieties that combine heat tolerance with the dark green trait [1,6].

The dark green color is not solely attributed to individual pigments but rather arises from the interactions among various pigment compounds, including chlorophyll, carotenoid, and anthocyanins [7]. Previous studies on green coloration in bunching onions have primarily focused on the roles of chlorophyll and carotenoids [8]; however, the molecular mechanisms underlying the dark green trait remain unclear. Additionally, leaf coloration has been reported to change in response to stress. Pheophytin, a compound formed under acidic or high-temperature conditions through chlorophyll degradation in tea leaves, is characterized by its brownish hue [9]. Anthocyanins have been observed to accumulate under drought conditions in wheat [10], and changes in the composition of phenolic compounds in olives (*Olea europaea* L.) have been shown to lead to black coloration [11]. Furthermore, sugars and proline function as osmolytes under abiotic stress conditions and have been reported to inhibit chlorophyll degradation in tomato and maize [12,13]. In addition, it has been reported that overexpression of the wax-related transcription factor *SHN1* in *Arabidopsis* results in the accumulation of cuticular wax in leaves, leading to a glossy and dark green phenotype [14]. Taken together, These findings suggest that leaf coloration in high-temperature summer cultivation is influenced not only by chlorophyll and carotenoids but also by the dynamics of pheophytin, anthocyanins, and phenolic compounds, as well as the involvement of sugars, proline, and waxes.

Therefore, to understand the mechanism behind the dark green coloration in bunching onions, it is necessary not only to measure pigment compounds but also to conduct metabolomic analysis targeting primary and secondary metabolites. Metabolomics enables a comprehensive analysis of biochemical states and stress-related compounds in plant cells, making it highly effective in identifying stress metabolism byproducts and compounds associated with adaptive responses [15,16]. In the *Allium* genus, metabolomics has been widely applied to analyze the effects of stress, cultivation conditions, drying methods, pathogen infection, and the introduction of alien chromosomes, contributing significantly to the identification of pigment compounds and metabolites [15,17,18,19,20].

Beyond understanding the factors contributing to the dark green trait, developing efficient evaluation methods is essential. Conventional chemical analyses are time-consuming and costly, highlighting the need for simpler approaches to estimating pigment compound contents. Spectral reflectance (400–700 nm) is a readily measurable property, and machine learning models utilizing these data have been shown to efficiently estimate pigment compound contents in various crops [21,22]. In the *Allium* genus, while studies using machine learning to estimate yield have been conducted for garlic and onion [23,24], no research has been conducted on predicting leaf blade color. These predictive models not only simplify the analysis of dark green coloration but also serve as valuable tools for plant monitoring and real-time diagnostics in agricultural practices.

In this study, we aimed to clarify the relationship between metabolome profiles and dark green coloration in bunching onion leaves and to evaluate the accuracy of predictive models for pigment compound estimation. To achieve this, qualitative and quantitative analyses of pheophytin *a* were conducted with high precision using an HPLC fluorescence detector, along with the quantitative analysis of chlorophyll *a*, chlorophyll *b*, lutein, and β-carotene and the evaluation of their optical properties. Additionally, widely targeted metabolite profiling based on targeted liquid chromatography–triple quadrupole mass spectrometry was performed. Finally, regression models based on spectral reflectance were developed to establish prediction equations for accurately estimating pigment compound contents. This study not only lays the foundation for improving the quality of bunching onions and developing stress-tolerant varieties but also opens avenues for future research focused on integrating genomic, transcriptomic, and metabolomic approaches to uncover the genetic basis of pigment accumulation and stress responses.

## 2. Materials and Methods

### 2.1. Materials and Growth Conditions

In this study, we examined two $F_{1}$ cultivars from Nakahara Seed Product Co., Ltd., Fukuoka, Japan― ‘Natsuhiko’ (NATS) and ‘Kaminari’ (KAMI); two purebred varieties from Yamaguchi Prefecture―‘YSG1go’ (YSG1) and ‘08S20-2’ (08S2); and three $F_{1}$ hybrid lines bred in Yamaguchi Prefecture (hereafter referred to as YB-$F_{1}$)―‘Yamahiko’ (YAMA), ‘Yamakou03’ (YAM3), and ‘2331’ (2331) (Appendix A). These plant materials were sown on 14 June 2023, and harvested on 21 August 2023, in the greenhouse of the Yamaguchi Prefectural Agriculture and Forestry General Technology Center (34° N, 131° E). The crops were sown in six rows on a ridge measuring 90 cm in width, with a spacing of 12 cm between rows and a seeding rate of 120 seeds per square meter. Nitrogen fertilizer was applied at a total rate of 1.0 kg per are (kg/a) over the course of the experiment. This was split into two applications: an initial 0.5 kg/a as a basal fertilizer at sowing and an additional 0.5 kg/a as topdressing at the two-leaf stage.

The watering conditions were as follows: On the sowing day, 48 L/$m^{2}$ of water was applied, with soil water tension maintained at 1.5 pF. During the germination phase (0–4 days), irrigation was conducted at a rate of 24 L/$m^{2}$ per application, keeping the pF value between 1.5 and 1.6. In the cotyledon stage (4–11 days), water was supplied every 2–3 days at 6 L/$m^{2}$, maintaining a pF range of 1.6–1.8. At the one-leaf stage (11–18 days), irrigation was performed every 2–3 days with 10 L/$m^{2}$ of water, keeping the pF value between 1.7 and 2.0. During the two-leaf stage (18–28 days), watering was increased to 12 L/$m^{2}$ daily, maintaining a pF range of 1.6–1.8. At the three-leaf stage (28–38 days), 6 L/$m^{2}$ of water was supplied every three days to maintain a pF value of 2.0–2.3. During the four-leaf stage (38–49 days), irrigation was carried out almost daily at a rate of 12 L/$m^{2}$, with the pF value maintained between 1.8 and 2.0. At the five-leaf stage (49–59 days), watering was conducted every three days at 5 L/$m^{2}$, keeping the pF level within 2.0–2.5. From the six-leaf stage (59 days onward), irrigation was adjusted as needed to maintain appropriate soil conditions.

After harvest, five plants were randomly selected from each variety and grouped into a single unit to ensure sufficient material for analysis. For example, in the case of YSG1, five such groups were prepared (YSG1-1 to YSG1-5), resulting in five biological replicates (n = 5). The number of biological replicates for each sample was as follows: NATS, n = 5; KAMI, n = 5; YSG1, n = 4; 08S2, n = 3; YAMA, n = 5; YAM3, n = 5; and 2331, n = 5. For all samples, outer and older leaves were removed, and the two youngest central leaves were selected for optical property measurements and chemical analyses.

Meteorological data, including air temperature (average, maximum, and minimum), relative humidity (average and minimum), and sunshine duration for Hofu City (34° N, 131° E) for the period from 13 June to 21 August 2019. https://www.data.jma.go.jp/stats/etrn/index.php, (accessed on 7 March 2025) (Appendix A).

### 2.2. Optical Property Measurements

As indicators of optical properties, the SPAD value and spectral reflectance (400–700 nm at 20 nm intervals) were measured. The SPAD value was determined using a chlorophyll meter (SPAD-502Plus, KONICA MINOLTA, INC., Tokyo, Japan), whereas spectral reflectance was measured with a spectrophotometer (NF555, Nippon Denshoku Industries Co., Ltd., Tokyo, Japan) [25]. For the spectrophotometer, reflectance correction was performed using black and white reference standards to minimize instrumental noise and improve the accuracy of spectral measurements. For optical property measurements, three plants were randomly selected from each biological replicate of each variety. The final value for each parameter was calculated by averaging the measurements taken from the central part of the first and second leaves.

### 2.3. Powdering of Samples

The samples were frozen at −80 °C after measuring their optical properties. The frozen samples were subsequently subjected to freeze-drying for three days using a TAITEC VD-250R freeze dryer with a vacuum pump (TAITEC, Saitama, Japan). The freeze-dried samples were ground using a small blender.

### 2.4. Pigment Compound Measurement

The acetone extraction method was established based on the approach described by Dissanayake et al. [26]. A precise amount of 20 mg of dried leaves was weighed and placed into a 15 mL tube, followed by the addition of 2.5 mL of chilled acetone. The mixture was vortexed for 2 min and subsequently subjected to ultrasonic treatment for 20 min under cool conditions. After centrifugation at 5000 rpm for 5 min at 10 °C, the supernatant was collected. The remaining residue was then extracted again by adding another 2.5 mL of chilled acetone, repeating the same procedure to obtain the supernatant. All collected supernatants were filtered through a 0.45 μm filter (Advantec, Tokyo, Japan) and used as sample solutions for the measurement of pheophytin *a* and pigment compounds (chlorophyll *a*, chlorophyll *b* and lutein, and β-carotene). The sample solutions were kept in a freezer at −20 °C and analyzed within three days.

The pheophytin *a* measurement method was established based on the approach described by Almela et al. [27]. Pheophytin *a* was measured by HPLC (Alliance e2695, Waters Corporation, MA, USA) equipped with a fluorescence detector (Waters 2475, Waters Corporation, MA, USA). The sample solutio was separated using a LiChroCART 250-4.0 Lichrospher 100 RP-18 (5 μm) column (KANTO CHEMICAL, Tokyo, Japan) with gradient elution consisting of two solvents: (A) 80% methanol solution (HPLC-grade) and (B) isopropanol (HPLC-grade). The elution program was as follows: (i) the initial condition was set to 100% (A), (ii) a linear gradient over 20 min transitioning to 40% (A) and 60% (B), and (iii) 40% (A) and 60% (B) maintained for 30 min. The flow rate was set at 1.0 mL/min, with the column temperature maintained at 30 °C and an injection volume of 10 μL. Fluorescence detection was carried out with excitation at 410 nm and emission at 660 nm. The pheophytin a standard was prepared by acidifying a chlorophyll *a* solution (dissolved in acetone) with a few drops of 0.1 N hydrochloric acid, followed by neutralization with an equivalent amount of 0.1 N sodium hydroxide.

The pigment compound measurement method was established based on the approach described by Dissanayake et al. [26], as previously cited for the acetone extraction method. Chlorophyll *a*, chlorophyll *b*, Lutein, and β-carotene were measured using an HPLC instrument (HPLC L-7000 series, HITACHI, Tokyo, Japan) equipped with a UV–Vis detector (HITACHI L7420, HITACHI, Tokyo, Japan) set to 435 nm for the detection of pigment compounds. The sample solution was separated using a LiChroCART 250-4.0 Lichrospher 100 RP-18 (5 μm) column (KANTO CHEMICAL, Tokyo, Japan) with gradient elution consisting of two solvents: (A) 80% methanol solution, prepared by mixing 400 mL of HPLC-grade methanol, 50 mL of ultrapure water, and 50 mL of 100 μM HEPES buffer (pH 7.5), and (B) an ethyl acetate. The gradient elution program was as follows: (i) the initial condition was 100% A; (ii) this was followed by a 20-min linear gradient to 50% A and 50% B; (iii) finally, 50% A and 50% B were held for 30 min. The flow rate was maintained at 1.0 mL/min, with a column temperature of 30 °C and an injection volume of 50 μL. The content of each pigment compound was calculated based on a calibration curve prepared using standard substances.

### 2.5. Metabolome Analysis

Sample preparation was automated using a liquid-handling system (Microlab STAR Plus, Hamilton Company, Reno, NV, USA) for dispensing, plate transfer, solvent evaporation (Ultravap Mistral, Porvair PLC, Norfolk, UK), dissolution, and filtration, following the method described by Sawada et al. [28]. A precisely weighed 4 mg portion of the powder was placed in a 2 mL tube along with a 5 mm zirconia ball (NIKKATO CORPORATION, Osaka, Japan). One milliliter of extraction solvent, consisting of 80% methanol and 0.1% formic acid, supplemented with 8.4 nmol/L lidocaine and 210 nmol/L 10-camphorsulfonic acid as internal standards, was then added to the tube, resulting in a final concentration of 4 mg/mL. Metabolite extraction was performed using a multi-bead shaker (Shake Master NEO, Biomedical Science, Tokyo, Japan) at 1000 rpm for 2 min.

Following centrifugation at 9100× *g* for 1 min, the supernatant was diluted fourfold with the extraction solvent, resulting in a final concentration of 1 mg/mL. Twenty-five microliters of each diluted sample was transferred to a 96-well plate, dried under a nitrogen stream, redissolved in 250 μL of ultrapure water (LC–MS grade), and filtered using a 0.45 μm pore size filter plate (Multiscreen HTS 384-Well HV, Merck, Rahway, NJ, USA). A 1 μL aliquot of the extract, adjusted to a final concentration of 100 ng/μL, was analyzed using LC-QqQ-MS for widely targeted metabolomics (UHPLC–Nexera MP/LCMS-8050, SHIMADZU, Tokyo, Japan). Three technical replicates were performed for each biological replicate. The LC-QqQ-MS parameters and MRM transitions were determined based on previous studies [29,30] and are summarized in Appendix A.

### 2.6. Statistical Methods

Data for SPAD value and total pigment compounds (chlorophyll *a*, chlorophyll *a* Pheophytin *a*, lutein, and β-carotene) were analyzed using one-way analysis of variance and Tukey’s multiple comparison test. A significance level of *p* < 0.05 was applied. Principal component analysis (PCA) was performed on spectral reflectance and pigment compounds to classify the varieties. The dataset was created by using the average values of 3 to 5 replicates for each sample and standardized with the StandardScaler function from the scikit-learn Python library. Subsequently, the PCA function from the same library was used to calculate the explained variance ratio for each principal component, and a scree plot was created. Based on the results, PCA was performed with two components. A random forest regression model was employed to predict each pigment compound content by using spectral reflectance from 400 nm to 700 nm at 20 nm intervals as input features. The dataset was split into training and test sets in an 80–20 ratio. Furthermore, to account for multicollinearity among input features, dimensionality reduction was performed using PCA. The number of components wasvaried while evaluating the coefficient of determination ($R^{2}$), and the optimal number of components was set to four, as it yielded the highest $R^{2}$. The model was implemented with the Scikit-learn library by using the following default hyperparameters: n_estimators = 100, max_depth = None, min_samples_split = 2, min_samples_leaf = 1, max_features = ‘sqrt’, and bootstrap = True. To ensure reproducibility, a fixed random seed (random_state = 42) was applied throughout the analysis. All analyses and visualizations were conducted by using Python 3.9.7, Pandas 1.3.4, Scikit-learn 0.24.2, and Matplotlib 3.4.3.

### 2.7. Classification of Green Color Patterns Using Metabolomics

The metabolomic analysis yielded intensity data for a total of 452 metabolites. The data were obtained with widely targeted metabolomics using liquid chromatography–tandem quadrupole mass spectrometry (Nexera MP with LCMS-8050, Shimadzu Corporation, Kyoto, Japan). Instrument variability and matrix effects were corrected by normalizing the sample peak area with the internal standard peak area. Missing values were replaced with a fixed value of 10, and the signal intensities of all samples (n = 3) were averaged. Metabolites with a signal-to-noise ratio (S/N), defined as the ratio of the average signal intensity to that of the extraction solvent control, below 5 across all samples were excluded from the analysis. Furthermore, metabolites with a relative standard deviation (RSD) exceeding 0.3 across all samples, as well as those with an RSD of 0, were excluded from the analysis. As a result of these filtering steps, a final dataset containing 186 metabolites was obtained. The data matrix was normalized to the median and auto-scaled. The processed dataset was then utilized for comparative analysis (Appendix A). First, partial least squares discriminant analysis (PLS-DA) was conducted on all 186 metabolites, and VIP scores were computed to evaluate overall trends in the metabolomic data among different green color types. Hierarchical clustering was applied to both samples and metabolites by using Ward’s method, with Euclidean distance as the similarity metric. Data preprocessing, including missing value imputation, S/N ratio filtering, RSD calculations, and intensity normalization to internal standards, was carried out using Python 3.9.7 and Pandas 1.3.4. Normalization and subsequent analyses were performed using MetaboAnalyst 6.0.

## 3. Results

### 3.1. Optical Property Measurements

The SPAD values and spectral reflectance measurements for all samples are presented in Appendix A. First, the average SPAD values of each variety and line were compared. The purebred variety 08S2 displayed the highest SPAD value, significantly exceeding those of the other varieties and lines. Similarly, the purebred variety YSG1 and the YB-$F_{1}$ lines YAM3 and 2331 showed relatively high SPAD values. In contrast, the $F_{1}$ cultivars NATS and KAMI showed significantly lower SPAD values than the others (Figure 1). These results suggest that the purebred varieties have darker green coloration, the $F_{1}$ cultivars have lighter green coloration, and the YB-$F_{1}$ lines exhibit intermediate characteristics.

### 3.2. Qualitative and Quantitative Determination of Pheophytin a and Pigment Compounds

Based on the HPLC analysis performed by using a fluorescence detector, we identified three peaks for the standard sample of pheophytin *a* (Figure 2A). Similarly, three peaks were also detected for chlorophyll *a*, which was used to prepare pheophytin *a*, suggesting that these peaks were influenced by components derived from chlorophyll *a* (Appendix A). Therefore, the largest peak at 43.045 min was identified as the peak for pheophytin *a*. A corresponding peak with the same retention time was also detected in the actual sample (YSG1) (Figure 2B). Based on this retention time, pheophytin *a* levels were quantified across all samples, with YSG1 exhibiting the highest levels (Figure 3). A comparison among different groups revealed that the $F_{1}$ cultivars generally had lower pheophytin *a* levels, the purebred varieties showed higher levels, and the YB-$F_{1}$ lines displayed intermediate levels. The quantification of pheophytin *a* has not been previously reported in bunching onion; however, compared to the results for *Allium ursinum*, it was found to be highly accumulated [31] Next, the results for chlorophyll *a*, chlorophyll *b* and lutein, β-carotene, and chlrophyll *a*/*b* ratio are provided as a Appendix A. The chlorophyll *a* and *b* values obtained in this study were lower than those previously reported for *Allium* species. However, it has been reported that chlorophyll accumulation in bunching onion is inhibited under high-temperature conditions [32], which may have influenced the results. These four pigment compounds, along with pheophytin *a*, were visualized as a stacked bar chart, and the total amount of the five pigment compounds was statistically analyzed using Tukey’s multiple comparison test. The purebred varieties 08S2 and YSG1, along with the YB-$F_{1}$ line 2331, exhibited significantly higher total pigment levels, showing the highest values compared to other varieties and lines, whereas the $F_{1}$ cultivars (NATS, KAMI) and the YB-$F_{1}$ line YAMA exhibited lower values, and the YB-$F_{1}$ line YAM3 showed intermediate values (Figure 4). Additionally, the analysis of the chlorophyll *a*/*b* ratio revealed that although no significant differences were observed among the cultivars and lines, KAMI exhibited the highest value (Appendix A). Furthermore, an analysis of the correlation between SPAD values and each pigment compound revealed the highest correlation with chlorophyll *a*, while the correlations with other pigment compounds were moderate (Table 1).

### 3.3. PCA Analysis Using Spectral Reflectance of Each Variety and Line

PCA was performed on pigment compound data for each variety and line. The principal component scores for each variety and line were plotted, and the results showed that the data were distributed around the center, with no clear separation observed (Appendix A). Next, PCA was conducted on the spectral reflectance data (400–700 nm) for each variety and line. The results showed that the data were projected onto Principal Component 1 (PC1) and Principal Component 2 (PC2), which together explained 99.6% of the total variance. The contribution rates of PC1 and PC2 were 85.5% and 14.1%, respectively (Figure 5A). The examination of the principal component loadings for PC1 and PC2 revealed that PC1 was composed of spectral reflectance, excluding the range of 540–580 nm. The positive side of PC1 indicated increases in reflectance in the violet, blue, yellow, and red regions, while the negative side indicated decreases in reflectance in these regions. PC2, on the other hand, was composed of spectral reflectance in the range of 540–580 nm, with increases in green spectral reflectance on the positive side and decreases on the negative side (Figure 5A). These findings suggest that PC1 differentiates dull green (grayish green) coloration based on increases in reflectance at the purple, blue, yellow, and red wavelengths, while PC2 distinguishes bright green and dark green based on variations in green spectral reflectance. By using the average PCA scores for each variety and line, the data were grouped into four clusters (Figure 5B). The $F_{1}$ cultivar KAMI was classified into Cluster 1 (green), the purebred variety 08S2 into Cluster 4 (gray green), and the YB-$F_{1}$ line 2331 into Cluster 2 (dark green). The remaining varieties and lines were grouped into Cluster 3, which exhibited intermediate characteristics based on PC1 and PC2. The average spectral reflectance curves in the visible light range for these green groups revealed that all groups exhibited the highest reflectance within the 540–560 nm range (Figure 6). Examining each group in detail, 08S2 displayed notably high reflectance across all wavelengths. KAMI showed slightly lower reflectance than 08S2 in the 540–560 nm range, but it exhibited the lowest reflectance in other wavelength regions. Finally, 2331 exhibited the lowest reflectance in the 540–560 nm range; however, its reflectance at other wavelengths was intermediate between those of 08S2 and KAMI.

### 3.4. Development of Prediction Models for Pigment Compounds Using Spectral Reflectance

The measured values of each pigment compound obtained in this study were used as response variables, and the spectral reflectance data (400–700 nm) were used as explanatory variables to construct prediction models. The spectral reflectance data (16 points from 400–700 nm) were subjected to PCA to reduce dimensionality and avoid multicollinearity issues, resulting in four principal components (PCs). These four PCs were used as independent variables, and random forest regression was applied to construct the prediction models. Given the limited sample size, Leave-One-Out Cross-Validation (LOOCV) was employed to assess the generalization performance of the models. As a result, the prediction model for chlorophyll *a* achieved coefficients of determination ($R^{2}$) and root mean square errors (RMSE) of 0.96 (RMSE = 2.28) in calibration, 0.88 (RMSE = 2.91) in testing, and 0.78 (RMSE = 5.28) in LOOCV, demonstrating a highly accurate regression model. Additionally, the scatter plot of predicted-versus-observed values confirmed that the data points were closely distributed along the ideal line (Table 2, Figure 7). On the other hand, the prediction model for β-carotene exhibited a high $R^{2}$ of 0.92 (RMSE = 0.19) in calibration but showed an $R^{2}$ of −0.10 (RMSE = 0.38) in the test phase, while in LOOCV, it yielded an $R^{2}$ of 0.60 (RMSE = 0.38), indicating moderate accuracy. These findings suggest a potential relationship between β-carotene and spectral reflectance. These results suggest a potential relationship between β-carotene and spectral reflectance. However, the low $R^{2}$ in the test data may have been influenced by the bias in data distribution. For the other pigment compounds, the $R^{2}$ values were low, confirming that the prediction accuracy was insufficient (Table 2).

### 3.5. Metabolomic Analysis of Three Green Color Patterns

Based on the PCA results, the samples were categorized into three distinct green color groups: green (KAMI, n = 3), dark green (2331, n = 5), and gray green (08S2, n = 5). Metabolomic analysis was performed after filtering the dataset from a total of 452 metabolites to 186 with data preprocessing. PLS-DA analysis revealed a clear separation among the three groups, with CP1 and CP2 contributing 21.5% and 13.8% of the variance, respectively. CP1 effectively distinguished samples into the green, dark green, and gray green groups (Figure 8A). Key contributors to this separation included amino acids such as alanine/sarcosine, norvaline/valine, and 5-aminovaleric acid. The green group exhibited higher amino acid accumulation than the other groups. Additionally, the dark green group showed elevated levels of quercetin derivatives and cyanidin derivatives. However, among the top 15 VIP scores, no specific metabolites were uniquely increased in the gray green group (Figure 8B). Next, the clustering of metabolites was performed, and a heatmap was created, revealing distinct accumulation patterns of metabolites across the three green color patterns (green, dark green, and gray green) (Figure 9). In the green group, metabolite accumulation was primarily associated with carbohydrates, including D-glucose-6-phosphate/D-fructose-6-phosphate, D-glucose-1-phosphate/D-galactose-1-phosphate, 1-kestose, and raffinose. In the dark green group, anthocyanins (e.g., cyanidin-3-glucoside) and their precursor, shikimic acid, were notably accumulated. In the gray green group, amino acids such as Gln and Asp, along with nicotinic acid mononucleotide, nicotinamide mononucleotide, phenylpropanoids, and sinapinic acid, were highly accumulated. These results indicate that each green color pattern activates different metabolic pathways. Specifically, sugar metabolism was emphasized in the green group, the pyridine nucleotide cycle and polyphenol metabolism were highlighted in the gray green group, and anthocyanin metabolism was prominent in the dark green group.

## 4. Discussion

### 4.1. Comparison of SPAD Values and Pigment Composition in Dark Green Varieties and Lines

This study provides valuable insights into the relationships among SPAD values, pigment compound contents, and green coloration patterns in heat-tolerant bunching onions. The comparison of SPAD values among the varieties and lines revealed distinct differences (Figure 1). The $F_{1}$ cultivars NATS and KAMI exhibited lower values, while the purebred varieties YSG1 and 08S2 showed higher values, with the YB-$F_{1}$ lines displaying intermediate values. While the quantification of five pigment compounds generally aligned with these trends, inconsistencies were observed. For example, YSG1 and 2331 had significantly lower SPAD values than 08S2, yet there were no notable differences in total pigment compound contents. Conversely, ‘YMA3’ showed SPAD values comparable to those of YSG1 and 2331 but had significantly lower total pigment levels. The correlation analysis between SPAD values and individual pigment compounds revealed a strong relationship with chlorophyll *a* but only moderate correlations with pheophytin *a*, β-carotene, and lutein (Table 1). This suggests that while the SPAD values effectively reflect chlorophyll *a* content, they are insufficient for assessing the more complex dark green coloration, which results from the interplay of multiple pigments. The limitations of SPAD meters arise from their reliance on transmittance measurements at specific wavelengths (650 nm and 940 nm), without accounting for other wavelengths critical to pigment interactions [33]. In sweet pepper (*Capsicum annuum* L.), it has been reported that considering parameters such as chroma and hue is necessary for a more accurate measurement of color [34]. Additionally, the purebred varieties exhibited a higher accumulation of total chlorophyll and pheophytin *a* compared to other varieties and lines (Appendix A). Since pheophytin is involved in photosynthetic electron transport, its accumulation may contribute to enhanced photosynthetic efficiency. However, it remains unclear how pheophytin accumulation is related to the progression of chlorophyll degradation. Therefore, further investigation is needed to elucidate the relationship between pheophytin and chlorophyll in detail.

### 4.2. Classification of Dark Green Varieties and Lines Using Spectral Reflectance

To classify varieties and lines based on the five pigment compounds, PCA was conducted. However, the principal component scores for each variety and line did not show clear differences (Appendix A). Subsequently, PCA was performed again on data on spectral reflectance in the visible light range. PC1 was associated with spectral reflectance at 400–700 nm, excluding the 540–560 nm range (Figure 5A). This component was interpreted as representing “dullness”, indicating variations in overall leaf brightness. In contrast, PC2 was specifically associated with the 540–560 nm range, where higher spectral reflectance corresponds to lighter green and lower reflectance indicates darker green [35]. Based on this, PC2 was interpreted as representing “brightness”. A plot of the principal component scores (PC1 × PC2) for each variety and line showed that KAMI, 08S2, and 2331 were classified into distinct clusters, corresponding to green, gray green, and dark green, respectively (Figure 5A,B).

### 4.3. Metabolic Profiles of the Three Green Color Groups

Metabolomic analysis was conducted by using these green color patterns. PLS-DA enabled clear classification into three green groups (Figure 8A), and heatmaps allowed us to identify metabolites that were specifically accumulated in each group (Figure 9). In the green group, 1-kestose, raffinose, trigonelline, and gamma-aminobutyric acid were accumulated. These metabolites have been shown to function as osmolytes in the *Allium* genus and *Arabidopsis* [36,37,38,39,40]. Heat stress suppresses chlorophyll synthesis. To adapt to such temperature fluctuations, osmolytes are accumulated, exerting protective effects [41]. In maize, it has been reported that under drought stress conditions, drought-tolerant lines accumulate more osmolytes than sensitive lines, thereby suppressing the reduction in chlorophyll *a* and contributing to the maintenance of a green color [42]. Moreover the chlorophyll *a*/*b* ratio are highly sensitive to the light environment and water stress, making them good indicators of oxidative stress in plant tissues [43]. In this study, only KAMI exhibited a high value, although not significantly different, suggesting that this may be involved in the stress response (Appendix A). Based on these findings, it is considered that the green group adapts to environmental stress by accumulating osmolytes and maintaining chlorophyll content.

In the gray green group, NAD intermediates such as nicotinic acid mononucleotide and β-nicotinamide mononucleotide, along with the Gln and Asp amino acids, were accumulated, indicating high activity of the pyridine nucleotide cycle. In this cycle, NAD and NADH function as cofactors in redox reactions and are believed to be directly involved in oxidative stress responses [44]. Additionally, the accumulation of sinapic acid, a phenolic compound, was observed. Sinapic acid, being an antioxidant and possessing the ability to effectively absorb ultraviolet radiation [45], is presumed to enhance photoprotective functions and contribute to environmental stress responses. Furthermore, spectral reflectance data confirmed that the gray green group exhibited higher overall reflectance than the other groups (Figure 6). In wheat and sorghum, it has been reported that wax-coated genotypes show an increase in reflectance in the visible light range [46,47]. Based on this, it is suggested that in the gray green group, the synthesis and accumulation of waxes in the cuticle layer may enhance light reflectance and protective functions against environmental stress. However, since waxes are lipids, they could not be clearly detected in the LC/MS-based metabolomic analysis conducted in this study. Therefore, future research should employ lipidomic analysis to further elucidate the characteristics and functional roles of waxes.

Finally, in the dark green group, it was confirmed that cyanidin glycosides and their precursor, shikimic acid, were accumulated (Figure 9). The typical UV–Vis spectrum of anthocyanins exhibits two major absorption clusters: one in the wavelength range of 260–280 nm and the other in the range of 490–550 nm [48]. Metabolomic analysis revealed that the dark green group (2331) accumulated more cyanidin than the other groups (Figure 9). Furthermore, the reflectance spectrum of 2331 showed a decrease in reflectance at 540–560 nm and an increase in reflectance at 640–700 nm (red wavelength region) compared with KAMI. These results are consistent with the absorption characteristics of cyanidin, suggesting that the presence of cyanidin contributes to the dark green appearance of leaves. Additionally, in Zikui Tea (*Camellia sinensis* cv. Zikui), it has been reported that the interaction between anthocyanins and chlorophyll results in a dark green appearance rather than a pure purple color, which aligns with the findings of this study [49]. Anthocyanins have been reported to accumulate in response to environmental stress, mitigating its effects by scavenging excess reactive oxygen species [50]. Similarly, in the case of 2331, it is considered that the accumulation of cyanidin in response to environmental stress may have influenced the development of a darker green color. In this study, we identified three distinct green color patterns: “green”, which indicates that the plant adapts to stress responses through osmotic adjustment; “gray green”, which indicates the accumulation of redox-related and phenolic compounds; and “dark green”, which is associated with anthocyanin accumulation. These differences are closely related to drought stress responses and hold great potential for agricultural applications. Future research should further elucidate the effects of drought stress and utilize these findings to develop drought-tolerant cultivars.

### 4.4. Construction of Prediction Models for Pigment Compounds Using Machine Learning

The validation results of the model based on LOOCV showed that the $R^{2}$ for chlorophyll *a* was 0.78, indicating high predictive accuracy. On the other hand, the $R^{2}$ value for β-carotene was 0.60, revealing some challenges in predictive accuracy (Table 2, Figure 7). For other pigment compounds, it was not possible to construct high-accuracy models (Table 2). Upon reviewing the dataset for the β-carotene regression model, we found that the β-carotene content was skewed toward a specific range, which may have hindered the model’s ability to learn effectively. As a countermeasure, it will be necessary to improve the regression model by increasing the sample size or implementing stratified sampling. Victor and Pozzobon [51] reported that measuring absorbance at 1 nm intervals within a wavelength range of 340–800 nm by using a spectrophotometer enabled highly accurate prediction of chlorophyll *a*, chlorophyll *b*, β-carotene, and lutein. These findings suggest that utilizing broader wavelength ranges, high-resolution spectral data, and their derivatives could lead to the development of more accurate predictive models.

Furthermore, the concentrations of carotenoids and chlorophyll provide essential information about the level of stress experienced by plants and their ability to tolerate it [52]. Therefore, highly accurate regression models for pigment compounds not only hold significance for developing selection markers for dark green traits but also suggest their potential as stress indicators. In this study, we constructed a method to rapidly and simply estimate chlorophyll *a* content by utilizing spectral reflectance data. In future research, improving accuracy for other pigment compounds is expected to further enhance practical applicability in agricultural fields and enable the use of these models as indicators for monitoring environmental stress and optimizing cultivation management. This study highlights the potential of combining physiological measurements, metabolomic analyses, and machine learning to improve our understanding of green coloration patterns and stress responses in bunching onions.

## 5. Conclusions

Building on these findings, several key directions for future research and applications can be considered. First, to elucidate the biochemical and genetic regulation of green coloration patterns, future studies should integrate genomics, transcriptomics, and metabolomics analyses. This approach would provide a more detailed understanding of the molecular mechanisms governing pigment biosynthesis and the regulation of responses to drought stress.

Next, improving the accuracy of the regression model requires increasing the sample size and utilizing high-resolution spectral reflectance data. For instance, as suggested by Victor and Pozzobon [51], incorporating absorbance data measured at 1 nm intervals can capture subtle absorption characteristics and interactions among pigments more precisely, thereby enhancing the robustness of the predictive model. This study demonstrated the usefulness of the predictive model under greenhouse conditions; however, for practical application in agricultural settings, its applicability must be evaluated, and its reproducibility under different field conditions must be verified through field trials. These trials will help clarify the impact of environmental factors on model accuracy and contribute to the development of more practical predictive methods. These research directions underscore the importance of integrating advanced technologies with applied agricultural research to develop innovative strategies for crop improvement under challenging environmental conditions.

## Figures and Tables

**Figure 1 metabolites-15-00226-f001:**
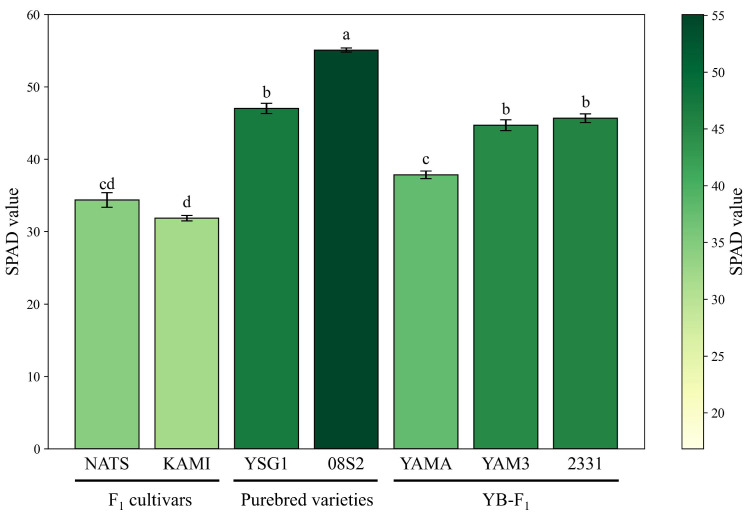
Bar chart of SPAD values across varieties and lines. Bars represent means ± standard errors. Different letters indicate significant differences at *p* < 0.05. The color gradient represents the SPAD values, with darker shades indicating higher values. NATS: ‘Natsuhiko’, KAMI: ‘Kaminari’, YSG1: ‘YSG1go’, 08S2: ‘08S20-2’, YAMA: ‘Yamahiko’, YAM3: ‘Yamakou03’, 2331: ‘2331’, Purebred varieties: Purebred varieties from Yamaguchi Prefecture, YB-$F_{1}$: $F_{1}$ hybrid lines bred in Yamaguchi prefecture.

**Figure 2 metabolites-15-00226-f002:**
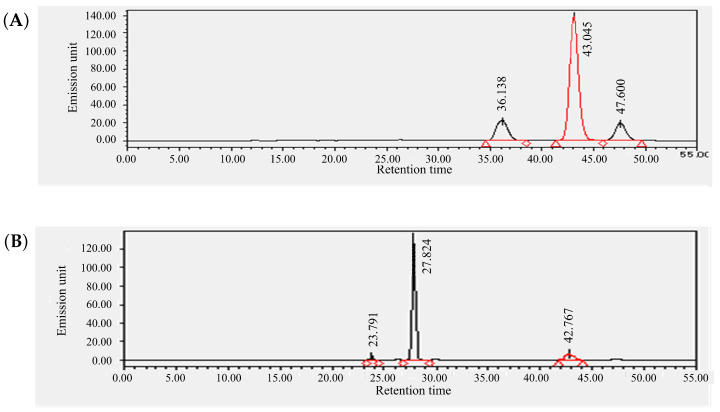
Chromatograms of pheophytin *a* (**A**) and YSG1 (**B**) obtained by using HPLC with a fluorescence detector.

**Figure 3 metabolites-15-00226-f003:**
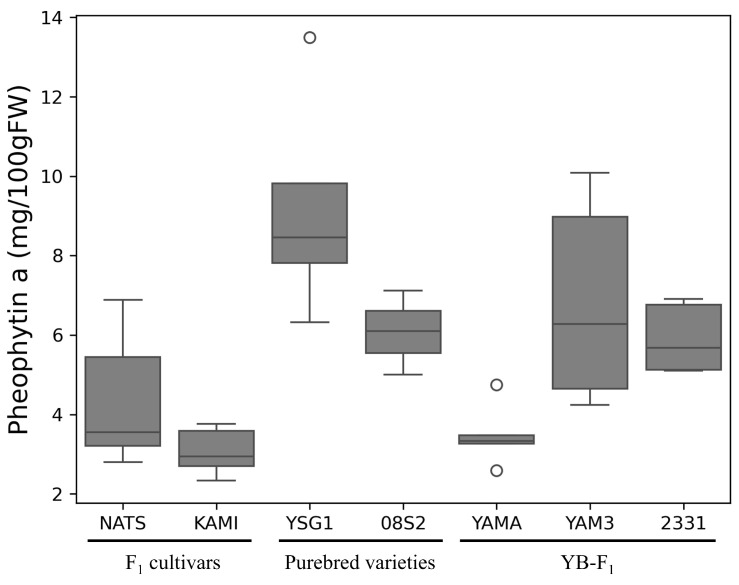
Boxplot of pheophytin *a* across varieties and lines. The box represents the interquartile range, the horizontal line within the box indicates the median, and the dots represent outliers.

**Figure 4 metabolites-15-00226-f004:**
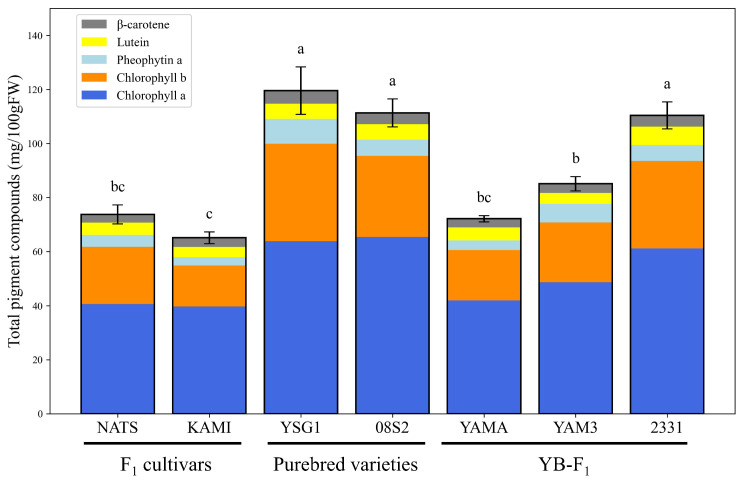
Bar chart of total pigment compound contents across varieties and lines. Different letters indicate significant differences at *p* < 0.05.

**Figure 5 metabolites-15-00226-f005:**
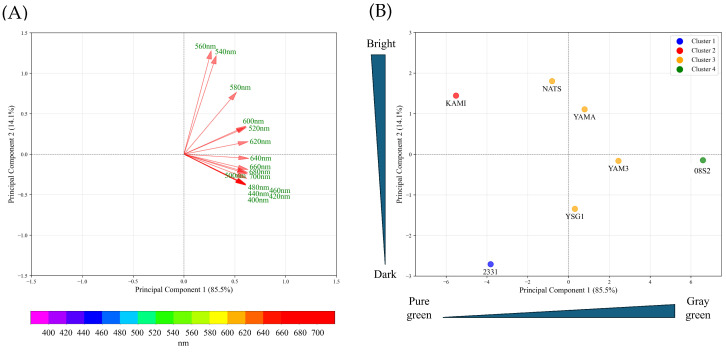
PCA loading plot (**A**) and PCA score plot with hierarchical clustering (**B**) of the dataset based on spectral reflectance. (**A**) Arrows represent the direction and magnitude of each variable’s influence, with longer arrows indicating stronger contributions. The color bar at the bottom represents the visible light spectrum. (**B**) The trend indicators on the left and bottom of the figure represent conceptual images of each principal component based on PCA loadings.

**Figure 6 metabolites-15-00226-f006:**
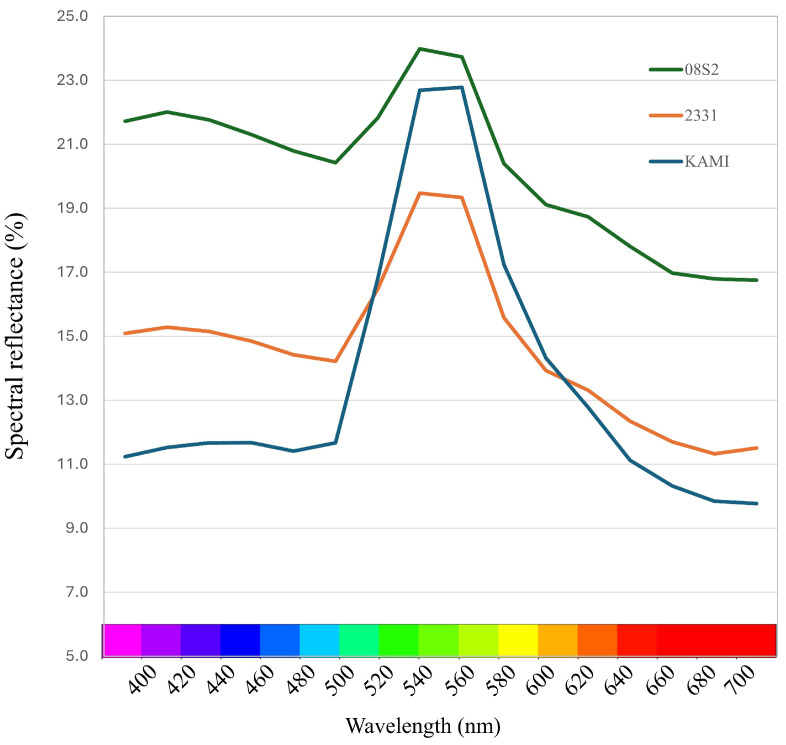
Average spectral reflectance curves (400–700 nm) for different green groups. The color bar at the bottom represents the visible light spectrum.

**Figure 7 metabolites-15-00226-f007:**
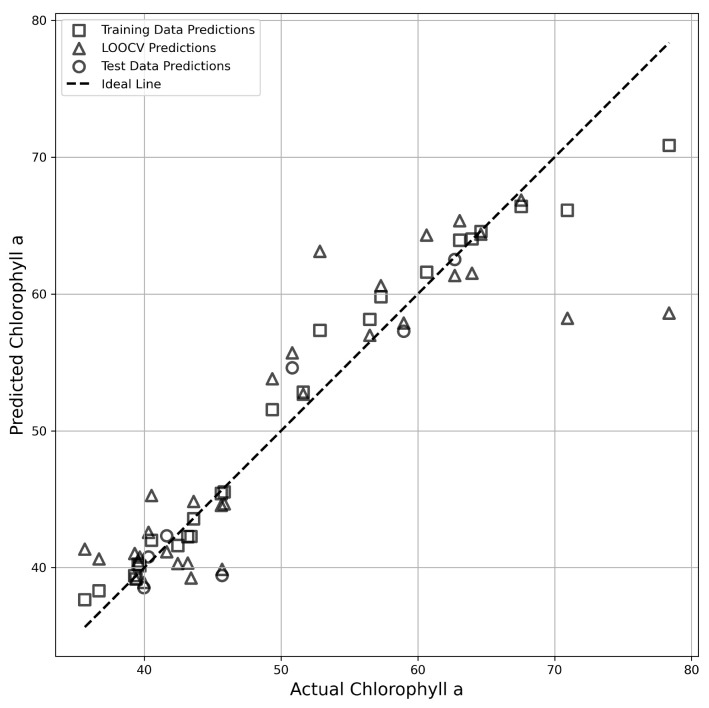
The graphical representation of measured vs. predicted values during the calibration, cross-validation, and testing steps for the models based on random forest for chlorophyll *a*.

**Figure 8 metabolites-15-00226-f008:**
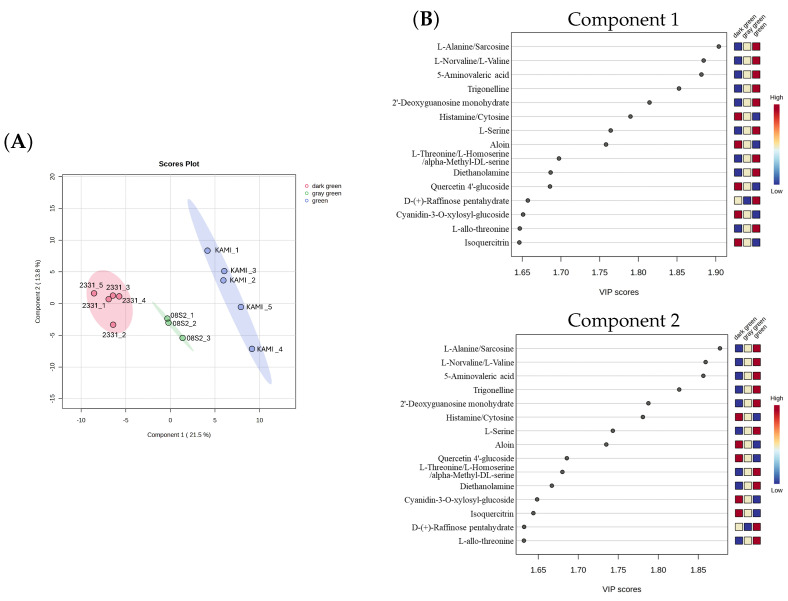
(**A**) PLS–DA (partial least squares–discriminant analysis) score plot and (**B**) VIP (Variable Importance in Projection) score plot, illustrating metabolite profiles based on different green groups. The contributions of metabolites to the first and second principal component axes are color-coded according to the contribution scale derived from the VIP scores.

**Figure 9 metabolites-15-00226-f009:**
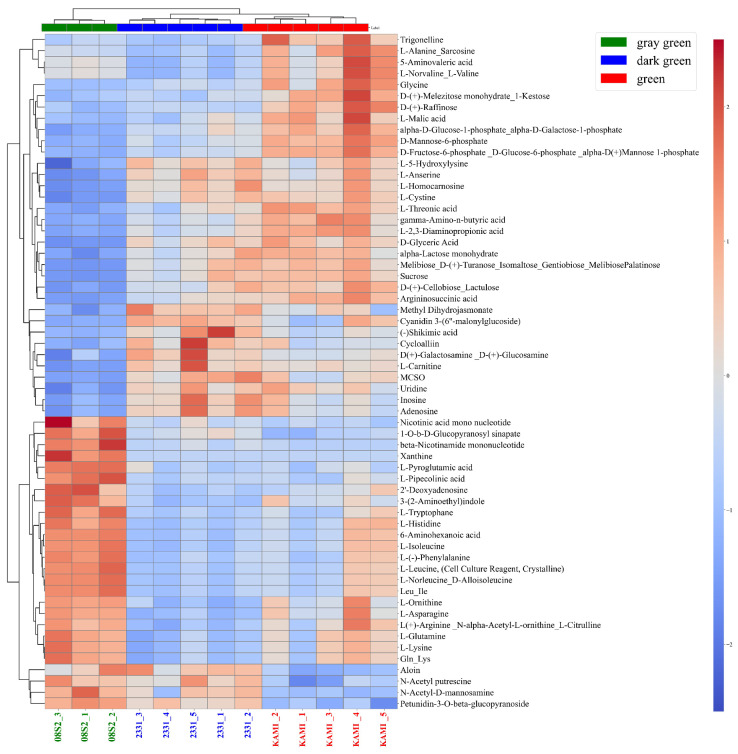
Dendrogram clustering and heatmap of metabolite profiles for the different green groups based on metabolome analysis.

**Table 1 metabolites-15-00226-t001:** Correlation coefficients (*r*) between SPAD values and each pigment compound.

Pigment Compound	Correlation Coefficient (*r*)
chlorophyll *a*	0.850
chlorophyll *b*	0.609
Pheophytin *a*	0.572
Lutein	0.405
β-Carotene	0.690

**Table 2 metabolites-15-00226-t002:** Quality parameters for the prediction of each pigment compound.

Pigment Compound	Quality Parameter	Calibration	Test	LOOCV
Chlorophyll *a*	$R^{2}$	0.96	0.88	0.78
	RMSE	2.28	2.91	5.28
Chlorophyll *b*	$R^{2}$	0.79	−1.07	0.38
	RMSE	4.50	6.70	7.08
Pheophytin *a*	$R^{2}$	0.86	0.47	0.17
	RMSE	0.96	1.58	2.26
Lutein	$R^{2}$	0.88	−0.57	−0.15
	RMSE	0.47	2.12	1.63
β-Carotene	$R^{2}$	0.92	−0.10	0.60
	RMSE	0.19	0.38	0.38

$R^{2}$: Coefficient of determination, RMSE: Root mean square error.

## Data Availability

The data used in the study were obtained from the websites of the Japan Meteorological Agency (https://www.data.jma.go.jp/stats/etrn/index.php, accessed on 7 March 2025) and the raw MS data can be downloaded from DROP Met database (https://prime.psc.riken.jp/menta.cgi/prime/drop_index#DM0069, accessed on 14 February 2025).

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
