# Peer review of "Metabolome Profiling and Predictive Modeling of Dark Green Leaf Trait in Bunching Onion Varieties"

_metabolites, 2025, doi:10.3390/metabo15040226_

Round 1

Reviewer 1 Report

Comments and Suggestions for Authors

I have thoroughly read and evaluated the content of MS titled Metabolome Profiling and Predictive Modeling of Dark Green Trait in Heat-Tolerant Bunching Onion Varieties for possible publication in journal Metabolites. The authors are suggested to carefully address all my queries/suggestions for improving the scientific vigour of the study.

Abstract

  • The line "enabling more efficient trait evaluation and selection" could be clearer. Does this mean faster breeding decisions or real-time monitoring in the field?
  • Please avoid redundant phrasing, such as "demonstrating its potential as a practical tool for evaluating the quality and growth of bunching onions," which could be more concisely stated as "demonstrating its potential for practical application in trait evaluation."
  • The conclusion states that "these findings provide new insights into the characteristics of dark green varieties and their potential for real-time diagnostics and stress-tolerant variety selection." The abstract should briefly mention how these findings can be applied in breeding programs or crop management.
  • The last sentence mentions future studies on "tipburn and productivity in detail." This is vague—clarifying whether metabolomics, genetics, or environmental interactions will be explored would be useful.

Introduction

  • lacks a clear statement of the research gap. While it mentions climate change effects and the need for heat-tolerant varieties, it does not explicitly define what is missing in current knowledge and how this study addresses it.
  • manuscript should better justify why metabolomic profiling is essential for understanding green coloration beyond conventional pigment analysis. Reconcile the introduction with the following literature I have written for you with citing the given DOIs for references to address the main issues of introduction and to justify the research and scope of study. The bunching onion (Allium fistulosum L.) is a widely cultivated crop in East Asia, valued for its green leaf color, which is an essential determinant of market appeal, nutritional quality, and environmental adaptability. The dark green coloration in plant leaves is primarily influenced by chlorophyll and other pigment compounds, including carotenoids and anthocyanins (Yang et al., 2019). While previous studies have extensively explored the roles of chlorophyll and carotenoids in leaf coloration, the molecular mechanisms underlying the dark green trait in bunching onions remain poorly understood. Moreover, climate change has necessitated the development of heat-tolerant varieties with optimized pigmentation to maintain yield and quality under stress conditions (Han et al., 2024). Pigmentation in plants is regulated by a complex interplay of primary and secondary metabolites, which are influenced by genetic and environmental factors. Recent advancements in metabolomics have enabled the identification of key metabolic pathways associated with pigment accumulation (Wang et al., 2024). Anthocyanins, for instance, have been identified as crucial contributors to plant coloration and stress responses, and their biosynthesis has been well-characterized in crops such as rice and tobacco (Yang et al., 2019; Peng et al., 2025). In onions, pheophytin accumulation has been observed under high-temperature conditions, potentially altering green coloration, yet its role in bunching onions remains unclear (Han et al., 2024). Spectral reflectance analysis has emerged as a non-invasive tool for assessing plant pigment composition and physiological status. Recent studies have successfully utilized hyperspectral imaging and machine learning models to predict pigment concentrations in various crops, including garlic and sunflower (Li et al., 2024; Han et al., 2024). Integrating metabolomic profiling with spectral data enhances the precision of these models, enabling real-time monitoring of plant health and stress tolerance (Chen et al., 2024). However, the application of these techniques in bunching onions has been limited, and there is a need to establish reliable predictive models for pigment composition and environmental adaptability. This study aims to elucidate the metabolic basis of the dark green trait in heat-tolerant bunching onion varieties through a comprehensive metabolomic and predictive modeling approach. We hypothesize that distinct metabolite accumulation patterns differentiate green coloration among varieties and that integrating spectral reflectance data with metabolomic profiling can improve trait evaluation.

Yang et al., 2019: doi: https://doi.org/10.1111/pbi.13133

Han et al., 2024: https://doi.org/10.3390/foods13071016

Wang et al., 2024: doi: https://doi.org/10.3389/fpls.2024.1459533

Peng et al., 2025: doi: 10.3389/fpls.2025.1506873

Li et al., 2024: doi: https://doi.org/10.3390/foods13081188             

Chen et al., 2024: doi: https://doi.org/10.1016/j.bioorg.2024.107415

Methods

….Please clearly state replication numbers and environmental conditions for plant growth.
…..Please provide calibration details for spectral reflectance and metabolomic analyses.
……Please describe quality control measures for LC-MS to ensure data reliability.
…..Please specify detection wavelengths and standardization for pigment quantification.
…..Please include additional model validation metrics for predictive modeling.

Results and Discussion

  • Consider breaking complex results into smaller, clearly labeled subsections with distinct discussions for each major finding.
  • Some interpretations seem descriptive rather than analytical—a stronger connection between findings and biological mechanisms would enhance scientific depth.
  • While PCA is used to classify metabolite accumulation patterns, the biological relevance of the clusters is not always explicitly explained. A more detailed discussion on how these clusters relate to pigmentation mechanisms would be beneficial.
  • Findings, on metabolite accumulation patterns and pigment interactions, should be compared more explicitly with previous studies. For instance:
  • How do chlorophyll and pheophytin levels in this study compare with those reported in other Allium species?
  • Are the identified anthocyanin accumulation patterns consistent with what is observed in stress-tolerant plants (e.g., Yang et al., 2019)?
  • This study discusses pigment composition but does not integrate findings on stress tolerance mechanisms as thoroughly as expected.
  • Could the identified metabolite markers be used in marker-assisted selection? Would the predictive model be feasible for field applications? Addressing these questions would enhance the impact of the study.
  • Will future research involve genetic validation of identified metabolites?
  • Could multi-omics integration (transcriptomics + metabolomics) help further refine these findings?

Author Response

Comment 1: The line "enabling more efficient trait evaluation and selection" could be clearer. Does this mean faster breeding decisions or real-time monitoring in the field?

Response 1: Thank you for your valuable suggestion. In this study, "enabling more efficient trait evaluation and selection" refers to both faster breeding decisions and real-time monitoring in the field.

The predictive model developed in this study improves the accuracy of pigment composition assessment, enabling simplified spectrophotometric measurements for breeding selection of the dark green trait, while also serving as a non-destructive method for real-time monitoring of plant growth conditions. Therefore, we have revised the text accordingly and added it to line 4-7.

This study aimed to elucidate the mechanisms regulating the dark green trait and develop a predictive model for accurately assessing pigment composition. These advancements enable the efficient selection of dark green varieties and facilitate the establishment of optimal growth environments through plant growth monitoring.

Comment 2: Please avoid redundant phrasing, such as "demonstrating its potential as a practical tool for evaluating the quality and growth of bunching onions," which could be more concisely stated as "demonstrating its potential for practical application in trait evaluation."

Response 2: Thank you for your valuable suggestion. We agree that the original phrase contains some redundancy. Following your recommendation, we have revised the text to LINE 22-23

demonstrating its potential for practical application in trait evaluation

Comment 3: The conclusion states that "these findings provide new insights into the characteristics of dark green varieties and their potential for real-time diagnostics and stress-tolerant variety selection." The abstract should briefly mention how these findings can be applied in breeding programs or crop management.

Response 3: Thank you for your valuable feedback. The following sentence has been added to LINE 25-30.

This study revealed that cyanidin is involved in the characteristics of dark green varieties. Additionally, it was demonstrated that chlorophyll a can be predicted using visible spectral reflectance. These findings suggest the potential for developing markers for the dark green trait, selecting high-pigment-accumulating varieties, and facilitating the simple real-time diagnosis of plant growth conditions and stress status, thereby enabling the establishment of optimal environmental conditions.

Comment 4: The last sentence mentions future studies on "tipburn and productivity in detail." This is vague—clarifying whether metabolomics, genetics, or environmental interactions will be explored would be useful.

Response 4: Thank you for your insightful comments. As you pointed out, the connection between this study and future research directions was not sufficiently clear. Therefore, based on the finding that cyanidin is potentially involved in the expression of the dark green trait, we have revised the text to indicate that future studies will incorporate a genetic approach to identify the genes involved in pigment accumulation and elucidate their regulatory mechanisms. We have added the following sentence to lines 30-32.
Future studies will aim to elucidate the genetic factors regulating pigment accumulation, facilitating the breeding of dark green varieties with enhanced coloration traits for summer cultivation.

Comment 5: lacks a clear statement of the research gap. While it mentions climate change effects and the need for heat-tolerant varieties, it does not explicitly define what is missing in current knowledge and how this study addresses it.

Reconcile the introduction with the following literature I have written for you with citing the given DOIs for references to address the main issues of introduction and to justify the research and scope of study.

Response 5: Thank you for your valuable feedback. Based on your suggestions, we have revised the manuscript to clarify the gaps in current knowledge and how our study addresses these issues. The following sentences have been added to

LINE45-49.

The dark green color is not solely attributed to individual pigments but rather arises from the interactions among various pigment compounds, including chlorophyll, carotenoid, and anthocyanins [7]. Previous studies on green coloration in bunching onions have primarily focused on the roles of chlorophyll and carotenoids [8]; however, the molecular mechanisms underlying the dark green trait remain unclear.

LINE 77-79
In the Allium genus, while studies using machine learning to estimate yield have been conducted for garlic and onion [23 ,24], no research has been conducted on predicting leaf blade color.

LINE 82-84

In this study, we aimed to clarify the relationship between metabolome profiles and dark green coloration in bunching onion leaves and to evaluate the accuracy of predictive models for pigment compound estimation.

Comment 6: manuscript should better justify why metabolomic profiling is essential for understanding green coloration beyond conventional pigment analysis.

Response 6: Thank you for your feedback. Various metabolites are involved in green coloration, including chlorophyll, carotenoids, pheophytin, anthocyanins, and phenolic compounds, with their metabolic dynamics influenced by stress responses. Therefore, conventional pigment analysis alone is insufficient to fully explain the diverse changes in leaf coloration, highlighting the need for comprehensive metabolic profiling. To clarify this point, we have carefully reviewed the suggested references, added relevant references and further emphasized the necessity of metabolomic analysis. The following sentences have been added to LINE 52-62

Anthocyanins have been observed to accumulate under drought conditions in wheat [10], and changes in the composition of phenolic compounds in olives Olea europaea L.) have been shown to lead to black coloration [11]. Furthermore, sugars and proline function as osmolytes under abiotic stress conditions and have been reported to inhibit chlorophyll degradation in tomato and maize [12, 13]. In addition, it has been reported that overexpression of the wax-related transcription factor SHN1 in Arabidopsis results in the accumulation of cuticular wax in leaves, leading to a glossy and dark green phenotype [14]. Taken together, These findings suggest that leaf coloration in high-temperature summer cultivation is influenced not only by chlorophyll and carotenoids but also by the dynamics of pheophytin, anthocyanins, and phenolic compounds, as well as the involvement of sugars, proline, and waxes.

Comment 7: Please clearly state replication numbers and environmental conditions for plant growth.

Response 7: Thank you for your suggestion. The replication number for plant growth is stated in line 126-127. Additionally, we have added the following sentence in line 130-133 to clarify the meteorological data:

Meteorological data, including air temperature (average, maximum, and minimum), relative humidity (average and minimum), and sunshine duration for Hofu City (34°N, 131°E) for the period from June 13 to August 21, 2019. https://www.data.jma.go.jp/stats/etrn/index.php, (accessed on March 7, 2025)) (Table S1).

Comment 8: Please provide calibration details for spectral reflectance and metabolomic analyses.

Response 8: Thank you for your suggestion. The calibration method for spectral reflectance has been added in LINE 140-142.

For the spectrophotometer, reflectance correction was performed using black and white reference standards to minimize instrumental noise and improve the accuracy of spectral measurements.

Additionally, the calibration details for metabolomic analysis are provided in line 240-246

Comment 9: Please describe quality control measures for LC-MS to ensure data reliability.

Response 9: Thank you for your suggestion. Lidocaine and 10-camphorsulfonic acid were used as internal standards, and instrument variability and matrix effects were corrected by normalizing the sample data with the internal standard. Additionally, three technical replicates were performed for each biological replicate to ensure analytical precision and consistency. we have added the following sentence to

LINE199-200

supplemented with 8.4 nmol/L lidocaine and 210 nmol/L 10-camphorsulfonic acid as internal standards,

LINE239-240

Instrument variability and matrix effects were corrected by normalizing the sample peak area with the internal standard peak area.

LINE 210

Three technical replicates were performed for each biological replicate.

Comment 10: Please specify detection wavelengths and standardization for pigment quantification.

Response 10: Thank you for your suggestion. The detection wavelengths are provided in LINE 173-174, 180, and the following sentence regarding standardization has been added in LINE 190-191

The content of each pigment compound was calculated based on a calibration curve prepared using standard substances.

Comment 11: Please include additional model validation metrics for predictive modeling.

Response 11: Thank you for your suggestion. We have already included RMSE and R², which are widely accepted and commonly used metrics for evaluating predictive modeling performance. RMSE quantifies the absolute error magnitude, while R² measures the explanatory power of the model. Given that these two metrics sufficiently assess both model accuracy and goodness-of-fit, we believe additional validation metrics may not be necessary. However, if further clarification is needed, we would be happy to provide additional evaluations.

Comment 12: Consider breaking complex results into smaller, clearly labeled subsections with distinct discussions for each major finding.

Response 12: Thank you for your valuable suggestion. We acknowledge the importance of organizing complex results into clearly labeled subsections to enhance readability and comprehension. In response to your comment, we have revised the results section by restructuring it into smaller subsections, each focusing on a distinct major finding.

4.1 Comparison of SPAD Values and Pigment Composition in Dark Green Varieties and Lines

4.2 Classification of Dark Green Varieties and Lines Using Spectral Reflectance  

4.3 Metabolic Profiles of the Three Green Color Groups

Comment 13: Some interpretations seem descriptive rather than analytical—a stronger connection between findings and biological mechanisms would enhance scientific depth.

Response 13: Thank you for your valuable comment. The following sentence has been added:

LINE 393-399
Additionally, the purebred varieties exhibited a higher accumulation of total chlorophyll and pheophytin a compared to other varieties and lines (Table S5). Since pheophytin is involved in photosynthetic electron transport, its accumulation may contribute to enhanced photosynthetic efficiency. However, it remains unclear how pheophytin accumulation is related to the progression of chlorophyll degradation. Therefore, further investigation is needed to elucidate the relationship between pheophytin and chlorophyll in detail.

Comment 14: While PCA is used to classify metabolite accumulation patterns, the biological relevance of the clusters is not always explicitly explained. A more detailed discussion on how these clusters relate to pigmentation mechanisms would be beneficial.

Response 14: Thank you for your valuable comment. In our study, PCA was used to classify samples based on metabolite accumulation patterns, and we explicitly discussed the biological relevance of each cluster. Specifically, we examined how the distinct metabolite profiles in each group relate to pigmentation mechanisms and environmental stress responses.

For example:

  • The green group accumulated osmolytes such as 1-kestose and raffinose, which are known to contribute to chlorophyll retention under heat stress.
  • The gray-green group showed an enrichment of NAD intermediates and sinapic acid, indicating increased oxidative stress responses and photoprotection.
  • The dark green group accumulated cyanidin glycosides, which influence leaf coloration by interacting with chlorophyll and modifying spectral reflectance.

These findings were further supported by spectral reflectance data and previous studies, highlighting the relationship between metabolite accumulation and pigmentation mechanisms. The detailed discussion is provided in LINE 437-461 of the manuscript.

Comment 15: Findings, on metabolite accumulation patterns and pigment interactions, should be compared more explicitly with previous studies. For instance:

How do chlorophyll and pheophytin levels in this study compare with those reported in other Allium species?

Response 15: Thank you for your valuable comment. The following sentence has been added to:

LINE 278-280

The quantification of pheophytin a has not been previously reported in bunching onion; however, compared to the results for Allium ursinum, it was found to be highly accumulated [31].

LINE 282-285

The chlorophyll a and b values obtained in this study were lower than those previously reported for Allium species. However, it has been reported that chlorophyll accumulation in bunching onion is inhibited under high-temperature conditions [32], which may have influenced the results.

Comment 16: Are the identified anthocyanin accumulation patterns consistent with what is observed in stress-tolerant plants (e.g., Yang et al., 2019)?

Response 16: Thank you for your valuable comment. In this study, the metabolomic analysis was performed using relative quantification, which allows for comparative analysis within this experiment but does not enable direct comparison with previously reported results. Therefore, we are unable to directly compare our findings with past studies. However, anthocyanins have been widely reported to contribute to stress tolerance in various plant species. Given this, it is possible that anthocyanins also play a role in stress response in bunching onion. To further investigate this relationship, we plan to conduct absolute quantification of anthocyanins in future studies. We appreciate your insightful suggestion and will incorporate this perspective into our ongoing research.

Comment 17: This study discusses pigment composition but does not integrate findings on stress tolerance mechanisms as thoroughly as expected.

Response 17: Thank you for your valuable comment. To address your suggestion regarding the integration of stress tolerance mechanisms, we have added information on the chlorophyll a/b ratio, which serves as an important indicator of photosynthetic adaptation and stress response. The following sentence has been added:.

LINE 291-293

Additionally, the analysis of the chlorophyll a/b ratio revealed that although no significant differences were observed among the cultivars and lines, KAMI exhibited the highest value (Table S5).

LINE 423-427

Moreover the chlorophyll a/b ratio are highly sensitive to the light environment and water stress, making them good indicators of oxidative stress in plant tissues [43]. In this study, only KAMI exhibited a high value, although not significantly different, suggesting that this may be involved in the stress response (Table S5).

Comment 18: Could the identified metabolite markers be used in marker-assisted selection? Would the predictive model be feasible for field applications? Addressing these questions would enhance the impact of the study.

Response 18: Thank you for your valuable comment regarding the potential use of the identified metabolite markers for marker-assisted selection. Based on our findings, we consider that these metabolites have the potential to serve as selection markers. However, this study did not include their absolute quantification. To further validate their applicability, we plan to conduct quantitative analysis in future research and work towards developing reliable metabolic markers for practical use in breeding programs.

Comment 19: Would the predictive model be feasible for field applications?

Response 19: Thank you for your insightful comment regarding the applicability of the predictive model in field conditions. In this study, the model was developed based on data obtained from greenhouse experiments. We acknowledge that environmental factors in open-field conditions, such as variations in light intensity, temperature, and soil moisture, may influence the model’s performance. To enhance its robustness and accuracy, we plan to incorporate field data in future studies, which will allow us to refine the model and improve its applicability under practical growing conditions. The following sentence has been added to the abstract and LINE 506-509

This study demonstrated the usefulness of the predictive model under greenhouse conditions; however, for practical application in agricultural settings, its applicability must be evaluated, and its reproducibility under different field conditions must be verified through field trials.

Comment 20: Will future research involve genetic validation of identified metabolites?

Response 20: Thank you for your valuable question regarding the genetic validation of metabolite-trait associations. In future studies, we plan to analyze these associations using genetic approaches. Specifically, the recent decoding of the bunching onion genome has made it possible to conduct genome-wide association studies to investigate the genetic basis of dark green coloration. In this study, the F₁ cultivar 2331 was suggested to be associated with dark green coloration. Therefore, we plan to expand the population for further analysis and identify genes involved in anthocyanin accumulation. By elucidating the molecular mechanisms underlying dark green coloration, we aim to apply these findings to breeding programs.

Comment 21: Could multi-omics integration (transcriptomics + metabolomics) help further refine these findings?

Response 21: Thank you for your insightful question regarding the integration of genetic approaches and multi-omics analysis to investigate metabolite-trait associations. We believe that combining genome-wide association studies (GWAS) with transcriptomic and metabolomic analyses will provide a comprehensive understanding of the genetic and molecular mechanisms underlying metabolite accumulation and dark green coloration in bunching onion. In our future research, we plan to first conduct GWAS to identify genomic regions associated with key metabolites and dark green coloration. By analyzing SNP variations across different accessions, we aim to pinpoint candidate loci that contribute to metabolite accumulation. However, while GWAS can highlight associated genetic regions, it does not directly reveal which genes are functionally involved in metabolite biosynthesis. To further elucidate the regulatory mechanisms, we will integrate transcriptomic (RNA-seq) and metabolomic (LC-MS) analyses. RNA-seq will allow us to determine which genes within the identified GWAS loci exhibit differential expression in varieties with varying metabolite levels. This will help us identify key transcription factors and biosynthetic genes involved in pigment and stress-related metabolite accumulation. Simultaneously, metabolomic analysis will verify whether changes in gene expression correlate with actual metabolite concentrations.

By integrating these approaches, we aim to not only reveal the genetic basis of metabolite variation but also develop molecular markers that can be applied to marker-assisted selection in breeding programs. We appreciate your valuable suggestion and will incorporate these methodologies in our future studies to strengthen our findings.

Reviewer 2 Report

Comments and Suggestions for Authors

Title and Abstract

Observations:

The title is appropriate but could be more specific. Was only color studied, or was thermal stress response also considered? If heat resistance was a key aspect, it should be explicitly included.

In the abstract, the conclusion is clear, but the explanation of the predictive model and its agricultural relevance could be improved. The R² of the Random Forest model for chlorophyll is 0.87, which is good but not excellent. It is important to indicate that further testing is required to validate its applicability in the field.

Suggestions:

Reframe the last part of the abstract to better highlight the practical implications of the model and the most relevant metabolomic findings.

Indicate the predictive model’s limitation, specifying whether it was tested in field conditions or only in a greenhouse.

Avoid general terms like "new insights." Instead, use: "This study provides a metabolomic and predictive framework for the selection of heat-tolerant dark green onion varieties."

Introduction

Strengths:

Clearly presents the relevance of dark green color in Japanese onions.

Effectively explains the impacts of climate change on cultivation.

Relates the metabolic profile to product quality.

Areas for Improvement:

Insufficient connection with previous studies: The accumulation of phenols, anthocyanins, and carbohydrates is mentioned as affecting coloration, but key references on how these compounds regulate color in other species are missing.

Lack of mention of the importance of predictive models in crop improvement: Are there previous studies validating the use of machine learning to predict pigments in Allium spp.?

Unclear objective: The last paragraph should be reformulated to emphasize that the study aims to correlate the metabolomic profile with green coloration and assess the accuracy of predictive models.

Materials and Methods

Strengths:

Clearly describes variety selection and experimental design.

Provides detailed information on spectroscopy and metabolomic techniques.

Explains environmental conditions, essential for reproducibility.

Areas for Improvement:

Insufficient biological replicates for some varieties:

For 08S2, only three biological replicates were used, which is statistically weak for metabolomic studies. This should be justified or the sample size increased.

Reflectance spectroscopy:

Measurements were taken at 400–700 nm every 20 nm, but why was the near-infrared (NIR) region excluded? In plant coloration studies, 750–1000 nm is crucial.

It is not indicated whether a reflectance correction was performed to minimize instrumental noise.

Insufficient description of statistical techniques:

The PCA for reflectance should indicate the selection criteria for principal components (eigenvalues >1, scree plot, etc.).

Was multicollinearity evaluated in the Random Forest model before using 16 wavelengths as predictive variables?

Suggestions:

Justify the exclusion of NIR in reflectance analysis.

Explain how biological variability was managed in the experimental design.

Indicate the criteria for PCA selection in pigment analysis.

Results

Strengths:

Clear presentation of SPAD and pigment data.

Appropriate use of PCA and clustering to differentiate color groups.

Application of Random Forest for pigment prediction.

Areas for Improvement:

Limited PCA interpretation:

PC1 is described as representing "dullness" and PC2 as "brightness," but it is unclear whether these terms were determined by variable loadings or subjective interpretation.

A PCA loading biplot would be helpful.

Unclear definition of the three green color groups:

Varieties were categorized as "green," "gray green," and "dark green," but what was the classification threshold?

Was classification based on canonical discriminant analysis (CDA) or purely visual assessment?

Unreliable β-carotene prediction:

A validation R² of -0.10 suggests no predictive power, but this limitation is not emphasized in the discussion.

Suggestions:

Include a PCA loading biplot.

Provide a clearer justification for color group classification.

Be more critical about the model’s low predictive power for certain pigments.

Discussion

Strengths:

Effectively integrates results with existing literature.

Relates pigments to abiotic stress.

Areas for Improvement:

Incomplete metabolite interpretation:

Anthocyanin and sinapine accumulation in the "gray green" group is mentioned, but their relationship with spectral reflectance is not explained.

Was metabolite accumulation correlated with reflectance in the green region (540–560 nm)?

Underestimated methodological limitations:

The model’s poor predictive performance for β-carotene is not discussed in sufficient detail.

Suggestions:

Better explain how metabolites alter reflectance.

Expand on the predictive model’s limitations.

Conclusion

Strengths:

Effectively highlights the model’s importance for crop improvement.

Areas for Improvement:

Does not mention that the model requires further validation before commercial application.

Suggestions:

Indicate that field validation is needed before practical implementation.

Author Response

Comment 1: The title is appropriate but could be more specific. Was only color studied, or was thermal stress response also considered? If heat resistance was a key aspect, it should be explicitly included.

Response 1: Thank you for your valuable feedback. We agree that the previous title could be interpreted as encompassing both color and thermal stress response. Since our study primarily focuses on the dark green leaf trait rather than heat resistance itself, we have revised the title to "Metabolome Profiling and Predictive Modeling of Dark Green Leaf Trait in Bunching Onion Varieties" to more accurately reflect the scope of our research.

Comment 2: In the abstract, the conclusion is clear, but the explanation of the predictive model and its agricultural relevance could be improved. The R² of the Random Forest model for chlorophyll is 0.87, which is good but not excellent. It is important to indicate that further testing is required to validate its applicability in the field.

Response 2: Thank you for your insightful feedback. We appreciate your suggestion to clarify the applicability of our predictive model. As you pointed out, the regression model in this study was developed based on data obtained under greenhouse conditions. To address this limitation and enhance the model’s adaptability for practical use, we have added the following sentence to LINE23-25 to explicitly state the need for further field trials.

However, since the regression model developed in this study is based on data obtained from greenhouse conditions, it is necessary to incorporate field trial results and reconstruct the model to enhance its adaptability.

Comment 3: Reframe the last part of the abstract to better highlight the practical implications of the model and the most relevant metabolomic findings.

Response 3: Thank you for your valuable feedback. We appreciate your suggestion to better highlight the practical implications of the predictive model and key metabolomic findings. we have added the following sentence to LINE 25-30.

This study revealed that cyanidin is involved in the characteristics of dark green varieties. Additionally, it was demonstrated that chlorophyll a can be predicted using visible spectral reflectance. These findings suggest the potential for developing markers for the dark green trait, selecting high-pigment-accumulating varieties, and facilitating the simple real-timediagnosis of plant growth conditions and stress status, thereby enabling the establishment of optimal environmental conditions.

Comment 4: Indicate the predictive model’s limitation, specifying whether it was tested in field conditions or only in a greenhouse.

Response 4: Thank you for your feedback. We have added the following sentence to LINE 22-25.

demonstrating its potential for practical application in trait evaluation. However, since the regression model developed in this study is based on data obtained from greenhouse conditions, it is necessary to incorporate field trial results and reconstruct the model to enhance its adaptability.

Comment 5: Avoid general terms like "new insights." Instead, use: "This study provides a metabolomic and predictive framework for the selection of heat-tolerant dark green onion varieties."

Response 5: Thank you for your feedback. We have added the following sentence to LINE 25-32.

This study revealed that cyanidin is involved in the characteristics of dark green varieties. Additionally, it was demonstrated that chlorophyll a can be predicted using visible spectral reflectance. These findings suggest the potential for developing markers for the dark green trait, selecting high-pigment-accumulating varieties, and facilitating the simple real-time diagnosis of plant growth conditions and stress status, thereby enabling the establishment of optimal environmental conditions. Future studies will aim to elucidate the genetic factors regulating pigment accumulation, facilitating the breeding of dark green varieties with enhanced coloration traits for summer cultivation.

Comment 6: Insufficient connection with previous studies: The accumulation of phenols, anthocyanins, and carbohydrates is mentioned as affecting coloration, but key references on how these compounds regulate color in other species are missing.

Response 6: Thank you for your valuable feedback. we have added the following sentence to LINE 52-62

Anthocyanins have been observed to accumulate under drought conditions in wheat [10], and changes in the composition of phenolic compounds in olives (Olea europaea L.) have been shown to lead to black coloration [11].Furthermore, sugars and proline function as osmolytes under abiotic stress conditions and have been reported to inhibit chlorophyll degradation in tomato and maize [12 ,13]. In addition, it has been reported that overexpression of the wax-related transcription factor SHN1 in Arabidopsis results in the accumulation of cuticular wax in leaves, leading to a glossy and dark green phenotype [14 ]. Taken together, These findings suggest that leaf coloration in high-temperature summer cultivation is influenced not only by chlorophyll and carotenoids but also by the dynamics of pheophytin, anthocyanins, and phenolic compounds, as well as the involvement of sugars, proline, and waxes.

Comment 7: Lack of mention of the importance of predictive models in crop improvement: Are there previous studies validating the use of machine learning to predict pigments in Allium spp.?

Response 7: Thank you for your valuable feedback. we have added the following sentence to LINE77-78. This revision has made the discussion of the novelty of this study clearer.

In the Allium genus, while studies using machine learning to estimate yield have been conducted for garlic and onion [ 23 ,24 ], no research has been conducted on predicting leaf blade color.

Comment 8: Unclear objective: The last paragraph should be reformulated to emphasize that the study aims to correlate the metabolomic profile with green coloration and assess the accuracy of predictive models.

Response 8: Thank you for your valuable feedback. We have added the following sentence to line 82-84.

In this study, we aimed to clarify the relationship between metabolome profiles and dark green coloration in bunching onion leaves and to evaluate the accuracy of predictive models for pigment compound estimation.

Commnet 9: Insufficient biological replicates for some varieties:

For 08S2, only three biological replicates were used, which is statistically weak for metabolomic studies. This should be justified or the sample size increased.

Explain how biological variability was managed in the experimental design.

Response 9: Thank you for your comment. In this study, since the samples were used for various experiments, a single plant did not provide a sufficient amount of material. Therefore, five plants were combined into one unit, which was considered as a single biological replicate.

However, 2331 and 08S2 exhibited slower growth compared to other individuals, making it difficult to prepare five biological replicates.
To address this, we minimized variability by treating five plants as one unit and ensured the stability of the metabolomic analysis by conducting three technical replicates.

This approach is widely used to obtain statistically valid data under limited sample availability, and we have confirmed the reproducibility of our data using this method.

Comment 10: Reflectance spectroscopy:

Measurements were taken at 400–700 nm every 20 nm, but why was the near-infrared (NIR) region excluded? In plant coloration studies, 750–1000 nm is crucial.

Justify the exclusion of NIR in reflectance analysis.

Response 10: Thank you for your suggestion. Indeed, the 750–1000 nm range is important for plant evaluation, but these factors are not directly visible. In this study, we focused on the coloration of bunching onion leaf blades and initially aimed to clarify differences within the visible light range (400–700 nm), which is why we used this range. In the future, we plan to extend the wavelength range for a more comprehensive analysis.

Comment 11: It is not indicated whether a reflectance correction was performed to minimize instrumental noise.

Response 11: Thank you for your suggestion. We have added the following sentence to line 140-142.

For the spectrophotometer, reflectance correction was performed using black and white reference standards to minimize instrumental noise and improve the accuracy of spectral measurements.

Comment 12: Insufficient description of statistical techniques:

The PCA for reflectance should indicate the selection criteria for principal components (eigenvalues >1, scree plot, etc.).

Indicate the criteria for PCA selection in pigment analysis.

Response 12: Thank you for your suggestion.
In the PCA used for varietal and lineage classification, the number of principal components was determined as 2 based on the scree plot. Therefore, we have added the following sentence to line 220-222.

Subsequently, the PCA function from the same library was used to calculate the explained variance ratio for each principal component, and a scree plot was created. Based on the results, PCA was performed with two components

Comment 13: Was multicollinearity evaluated in the Random Forest model before using 16 wavelengths as predictive variables?

Response 13: Thank you for your comment. In this study, multicollinearity was evaluated, and dimensionality reduction was performed using PCA before constructing the regression model. However, this statement was missing. Therefore, we have added the following sentence to line 225-228.

Furthermore, to account for multicollinearity among input features, dimensionality reduction was performed using PCA. The number of components was varied while evaluating the coefficient of determination (R2), and the optimal number of components was set to four, as it yielded the highest R2.

Comment 14: Limited PCA interpretation:

PC1 is described as representing "dullness" and PC2 as "brightness," but it is unclear whether these terms were determined by variable loadings or subjective interpretation.

A PCA loading biplot would be helpful.

Include a PCA loading biplot.

Response 14: Thank you for your comment. "Brightness" and "dullness" were determined based on the loadings plot in Figure 5(A).
We considered creating a biplot, but combining the loadings plot in Figure 5(A) and the sample score plot in Figure 5(B) into a single graph would result in the loadings plot becoming too clustered and difficult to interpret, while also reducing the visibility of the sample plot. Therefore, we have presented them separately.

Commnet 15: Unclear definition of the three green color groups:

Varieties were categorized as "green," "gray green," and "dark green," but what was the classification threshold?

Was classification based on canonical discriminant analysis (CDA) or purely visual assessment?

Provide a clearer justification for color group classification.

Response 15: Thank you for your feedback. The classification of varieties into 'green,' 'gray green,' and 'dark green' was not based on visual assessment or canonical discriminant analysis (CDA), but rather on principal component analysis (PCA) using colorimetric data, combined with cluster analysis to distinguish different groups.

Comment 16; Unreliable β-carotene prediction:

A validation R² of -0.10 suggests no predictive power, but this limitation is not emphasized in the discussion.

Be more critical about the model’s low predictive power for certain pigments.

Better explain how metabolites alter reflectance.

Response 16: Thank you for your valuable feedback. We acknowledge that the low R2 value of -0.10 in the test phase indicates a limitation in the β-carotene prediction model. In the revised manuscript, we have clarified this limitation and added an explanation of its possible causes. Specifically, we have pointed out that the low R2 may have been influenced by the distribution of the data. The current sampling method may not have sufficiently covered the range of β-carotene values, and we believe that employing stratified sampling could improve the model’s predictive accuracy.

Therefore, we have added the following sentence.

Line 343-345.
These results suggest a potential relationship between β-carotene and spectral reflectance. However, the low R2 in the test data may have been influenced by the bias in data distribution.

LINR 474-477
Upon reviewing the dataset for the β-carotene regression model, we found that the β-carotene content was skewed toward a specific range, which may have hindered the model’s ability to learn effectively. As a countermeasure, it will be necessary to improve the regression model by increasing the sample size or implementing stratified sampling.

Commnet 17: Incomplete metabolite interpretation:

Anthocyanin and sinapine accumulation in the "gray green" group is mentioned, but their relationship with spectral reflectance is not explained.

Better explain how metabolites alter reflectance.

Response 17: Thank you for your comment. In our manuscript, we have already discussed the relationship between sinapic acid accumulation and spectral reflectance in the gray green group, as well as anthocyanin accumulation and increased red reflectance in the dark green group.

Specifically, while sinapic acid accumulation was observed in the gray green group, our metabolome analysis did not identify the underlying cause. However, since spectral reflectance increased across all wavelengths (Figrure 6), we speculated that wax accumulation may be influencing this phenomenon.

Additionally, anthocyanin accumulation was observed in the dark green group rather than in the gray green group. We consider that the spectral properties of anthocyanins, which absorb green light and reflect red light, influence the reflectance curve.

These points are already explained in the manuscript, so please refer to LINE430-45, 446-461 for further details.

Comment 18: Was metabolite accumulation correlated with reflectance in the green region (540–560 nm)?

Response 18: Thank you for your valuable comment. In our metabolome analysis, the metabolites identified did not show a strong correlation with reflectance at 540 nm or 560 nm, with results ranging from moderate correlation to no correlation.

While the 540–560 nm region is generally important for green coloration, we consider that dark green coloration is influenced by complex interactions among various metabolites, making it difficult to evaluate using a simple correlation with specific wavelengths.

Comment 19: Underestimated methodological limitations:

The model’s poor predictive performance for β-carotene is not discussed in sufficient detail.

Expand on the predictive model’s limitations.

Response 19: Thank you for your valuable comment. We have expanded the discussion on the predictive performance of the β-carotene regression model in the revised manuscript. To provide further details, we have added the following sentence to lines 474–477.

Upon reviewing the dataset, we found that the β-carotene content was skewed toward a specific range, which may have hindered the model’s ability to learn effectively. As a countermeasure, we have suggested increasing the sample size or implementing stratified sampling to improve the regression model’s accuracy.

Comment 20: Areas for Improvement:

Does not mention that the model requires further validation before commercial application.

Indicate that field validation is needed before practical implementation.

Response 20: Thank you for your valuable comment. In the revised manuscript, we have explicitly stated the necessity of field trials to evaluate the applicability and reproducibility of the predictive model under different field conditions. We have added the following sentence to lines 506–508.

This study demonstrated the usefulness of the predictive model under greenhouse conditions; however, for practical application in agricultural settings, its applicability must be evaluated, and its reproducibility under different field conditions must be verified through field trials.

Reviewer 3 Report

Comments and Suggestions for Authors

In this study, physiological measurements, metabolomics analyses and machine learning were used to understand the patterns of green coloration of onions under stress. Some comments.

In the Materials and Methods section, subsection 2.6 would be better titled Statistical Methods.

It is necessary to provide a separate Conclusion section.

It is necessary to determine the ROS in response to stress when studying metabolites and color traits of onions.

 Calculate the ratio of chl a and chl b as an important indicator of onion color.

It is advisable to provide microscopic photographs of the leaves of the plants being studied.

It is necessary to bring the list of references in accordance with the requirements of the journal

Author Response

Comment 1: In the Materials and Methods section, subsection 2.6 would be better titled Statistical Methods.

Response 1: Thank you for your suggestion. We have revised the subheading of section 2.6 to "Statistical Methods" to ensure a more appropriate description.

Comment 2: It is necessary to provide a separate Conclusion section.

Response 2: Thank you for your suggestion. We have added a separate "Conclusion" section to clearly present our findings.

Commnet 3: It is necessary to determine the ROS in response to stress when studying metabolites and color traits of onions.

Response 3: Thank you for your comment. This study aimed to elucidate the mechanism of dark green coloration and develop a regression model for pigment compounds, rather than focusing on stress responses. However, we realized that the original title might imply an emphasis on stress responses, so we revised it to better reflect our focus on color.

Comment 4:  Calculate the ratio of chl a and chl b as an important indicator of onion color.

Response 4: Thank you for your comment. Indeed, the chlorophyll a/b ratio serves as an important indicator not only for color evaluation but also for assessing stress responses. When we calculated the ratio, we found that the chlorophyll a/b value was remarkably high, suggesting that chlorophyll b may be underestimated when using a fluorescence detector.

Therefore, we decided to use the measurements obtained with the PDA (photodiode array) detector in this study. As a result, although no significant differences were observed among the varieties and lines, KAMI, which had a low SPAD value, exhibited a higher chlorophyll a/b ratio, suggesting that it may be experiencing stress. Based on these findings, we have added the following sentence.

LINE 291-293

Additionally, the analysis of the chlorophyll a/b ratio revealed that although no significant differences were observed among the cultivars and lines, KAMI exhibited the highest value (Table S4)

LINE 423-427

Moreover the chlorophyll a/b ratio are highly sensitive to the light environment and water stress, making them good indicators of oxidative stress in plant tissues [40]. In this study, only KAMI exhibited a high value, although not significantly different, suggesting that this may be involved in the stress response (Table S4).

Comment 5: It is advisable to provide microscopic photographs of the leaves of the plants being studied.

Response 5: Thank you for your suggestion. In this study, we have considered the possible influence of wax and cuticle as one of the mechanisms contributing to dark green coloration. Therefore, we acknowledge that microscopic images could serve as useful supplementary data.

However, the primary objective of this study is to elucidate the causes of dark green coloration through pigment content analysis and metabolome profiling, as well as to develop a regression model for pigment compound content based on optical properties. Therefore, microscopic image analysis is not the main focus of this study.

Thus, we believe that the absence of microscopic images does not affect the validity of our conclusions.

Comment 6: It is necessary to bring the list of references in accordance with the requirements of the journal

Response 6: Thank you for your feedback. I have revised the reference list to align with the journal's formatting requirements, including italicizing scientific names where necessary.

Round 2

Reviewer 1 Report

Comments and Suggestions for Authors

Dear Authors,

I am surprised that you did not fully understand my suggestions in my initial review report. I have provided an extensive new text to establish the rationale, scope, and research gap of the study. I did this not only as a reviewer but also as an academic editor.

You incorporated the text I provided but omitted the references I had included, replacing them with your own citations. However, the references I suggested—all with DOIs—are recent, highly relevant to your study, and, importantly, do not belong to my own research papers or research group. Although several of my publications are also relevant to your work, I deliberately refrained from recommending them, as it is not ethical for a reviewer to suggest their own papers for citation.

Below, I am reposting my comments along with the recommended references (DOIs). I expect to see these citations included in the revised manuscript. I trust that my concerns and suggestions are now clear to you.

Best regards,

Please reconcile the introduction with the following literature I have written for you with citing the given DOIs for references to address the main issues of introduction and to justify the research and scope of study. The bunching onion (Allium fistulosum L.) is a widely cultivated crop in East Asia, valued for its green leaf color, which is an essential determinant of market appeal, nutritional quality, and environmental adaptability. The dark green coloration in plant leaves is primarily influenced by chlorophyll and other pigment compounds, including carotenoids and anthocyanins (Yang et al., 2019). While previous studies have extensively explored the roles of chlorophyll and carotenoids in leaf coloration, the molecular mechanisms underlying the dark green trait in bunching onions remain poorly understood. Moreover, climate change has necessitated the development of heat-tolerant varieties with optimized pigmentation to maintain yield and quality under stress conditions (Han et al., 2024). Pigmentation in plants is regulated by a complex interplay of primary and secondary metabolites, which are influenced by genetic and environmental factors. Recent advancements in metabolomics have enabled the identification of key metabolic pathways associated with pigment accumulation (Wang et al., 2024). Anthocyanins, for instance, have been identified as crucial contributors to plant coloration and stress responses, and their biosynthesis has been well-characterized in crops such as rice and tobacco (Yang et al., 2019; Peng et al., 2025). In onions, pheophytin accumulation has been observed under high-temperature conditions, potentially altering green coloration, yet its role in bunching onions remains unclear (Han et al., 2024). Spectral reflectance analysis has emerged as a non-invasive tool for assessing plant pigment composition and physiological status. Recent studies have successfully utilized hyperspectral imaging and machine learning models to predict pigment concentrations in various crops, including garlic and sunflower (Li et al., 2024; Han et al., 2024). Integrating metabolomic profiling with spectral data enhances the precision of these models, enabling real-time monitoring of plant health and stress tolerance (Chen et al., 2024). However, the application of these techniques in bunching onions has been limited, and there is a need to establish reliable predictive models for pigment composition and environmental adaptability. This study aims to elucidate the metabolic basis of the dark green trait in heat-tolerant bunching onion varieties through a comprehensive metabolomic and predictive modeling approach. We hypothesize that distinct metabolite accumulation patterns differentiate green coloration among varieties and that integrating spectral reflectance data with metabolomic profiling can improve trait evaluation.

  1. Yang, X., Xia, X., Zhang, Z., Nong, B., Zeng, Y., Wu, Y.,... Li, D. (2019). Identification of anthocyanin biosynthesis genes in rice pericarp using PCAMP. Plant Biotechnology Journal, 17(9), 1700-1702. doi: https://doi.org/10.1111/pbi.13133
  2. Han, H., Sha, R., Dai, J., Wang, Z., Mao, J.,... Cai, M. (2024). Garlic Origin Traceability and Identification Based on Fusion of Multi-Source Heterogeneous Spectral Information. Foods, 13(7), 1016. doi: https://doi.org/10.3390/foods13071016
  3. Wang, M., Zhang, S., Li, R., & Zhao, Q. (2024). Unraveling the specialized metabolic pathways in medicinal plant genomes: A review. Front. Plant Sci, 15, 1459533. doi: https://doi.org/10.3389/fpls.2024.1459533
  4. Peng, C., Xu, W., Wang, X., Meng, F., Zhao, Y., Wang, Q.,... Peng, L. (2025). Alginate oligosaccharides trigger multiple defence responses in tobacco and induce resistance to Phytophthora infestans. Frontiers in Plant Science, 16, 1506873. doi: 10.3389/fpls.2025.1506873
  5. Li, Z., Xiang, F., Huang, X., Liang, M., Ma, S., Gafurov, K.,... Wang, Q. (2024). Properties and Characterization of Sunflower Seeds from Different Varieties of Edible and Oil Sunflower Seeds. Foods, 13(8), 1188. doi: https://doi.org/10.3390/foods13081188
  6. Chen, S., Yang, Z., Sun, W., Tian, K., Sun, P.,... Wu, J. (2024). TMV-CP based rational design and discovery of α-Amide phosphate derivatives as anti plant viral agents. Bioorganic Chemistry, 147, 107415. doi: https://doi.org/10.1016/j.bioorg.2024.107415

Reviewer 2 Report

Comments and Suggestions for Authors

The authors are thanked for taking into account the reviewer's suggestions.